# Nutritional Supplement with Fermented Soy in Patients Under Active Surveillance for Low-Risk or Intermediate-Risk Prostate Cancer: Results from the PRAEMUNE Trial

**DOI:** 10.3390/cancers16213634

**Published:** 2024-10-28

**Authors:** Hans Van der Eecken, Diederik De Cock, Eduard Roussel, Alexander Giesen, Bram Vansevenant, Lieven Goeman, Thierry Quackels, Steven Joniau

**Affiliations:** 1Department of Urology, University Hospital Brussels, 1090 Brussels, Belgium; hans.vandereecken@uzbrussel.be; 2Biostatistics and Medical Informatics Research Group, Department of Public Health, Faculty of Medicine and Pharmacy, Vrije Universiteit Brussel (VUB), 1050 Brussels, Belgium; diederik.de.cock@vub.be; 3Department of Urology, University Hospitals Leuven, 3000 Leuven, Belgium; eduard.roussel@uzleuven.be (E.R.); alexander.giesen@uzleuven.be (A.G.); bram.vansevenant@uzleuven.be (B.V.); 4Department of Urology, AZ Delta Roeselare, 8800 Roeselare, Belgium; lieven.goeman@azdelta.be; 5Department of Urology, Erasmus University Hospital Brussels, 1070 Brussels, Belgium; thierry.quackels@erasme.ulb.ac.be

**Keywords:** fermented soy, PSA, prostate cancer, active surveillance

## Abstract

Many beneficial mechanisms of action are known from soy isoflavones, including in prostate and prostate cancer. Previous research was able to demonstrate a clear effect of PSA modulation, and this could potentially lead to better selection of patients at increased risk of occult prostate carcinoma. In low-grade localized prostate cancer, more and more active surveillance is being proposed and a lot of research is being carried out to see if certain factors or treatments may also act in this regard. This formed the basis to study the effect of soy isoflavones in patients under active surveillance. In this study, we were able to observe PSA modulation in a large proportion of patients, resulting in fewer follow-up examinations and fewer positive control biopsies. Potentially, in active surveillance, a soy supplement could be useful to help select patients who could continue to be followed up or who should be treated earlier.

## 1. Introduction

There is a general consensus that active surveillance (AS) can be applied as standard treatment for patients with low-risk prostate cancer (PCa), and by extension in selected patients with (favorable) intermediate-risk PCa [1,2,3,4]. AS involves follow-up of diagnosed PCa to defer active treatment until the disease is reassessed as higher risk, for example, by an increase in International Society of Urological Pathology (ISUP) score or tumor stage [5].

AS has gained its position mainly because, on the one hand, active local treatments are associated with increased risks of treatment-related adverse events (TRAEs) regarding continence and potency with compromising health-related quality of life (HRQoL) [6]. On the other hand, we know that there is also overdiagnosis and overtreatment in PCa [7].

In a variety of different protocols in AS, efforts are being made to reach a general consensus, mainly to reduce burdensome consecutive prostate biopsies [8]. Crucial in AS are prostate-specific antigen (PSA), digital rectal examination (DRE), magnetic resonance imaging (MRI) and prostate biopsies. Control biopsies are considered necessary, at least after 2 years and at least every 3 years thereafter [9].

Recent consensus recommends that if both PSA kinetics and MRI are stable, control biopsies can be omitted [10]. However, a change in PSA or DRE should be an indication to proceed to MRI with or without biopsy, before immediately deciding on active treatment.

Given the importance of stabilizing PSA kinetics in AS, it is highly worth focusing on this. Previous research by our team showed, first in a retrospective setting, and then in a prospective setting in patients with previous negative biopsies, that administration of a dietary supplement containing fermented soy, can provide a modulating or stabilizing effect on the PSA [11,12]. Oncological results in the prospective study showed that a modulating effect on PSA seemed to influence clinical practice and the number of PCa-related investigations, and that prescribing a fermented soy supplement in patients at increased risk of PCa could lead to a better selection of patients at real increased risk of having occult PCa.

This study investigated the effect of an 18-month nutritional supplement challenge in patients under AS for low- and intermediate-risk PCa. The purpose was mainly through evaluation of PSA progression to see if we can support a hypothesis that a fermented soy supplement can help select patients who can remain under AS or who should switch to active treatment.

## 2. Materials and Methods

### 2.1. Patients

Inclusion criteria were patients after positive biopsies eligible for AS: low-risk patients with ISUP grade 1, clinical stage cT1 or cT2a, PSA < 10 ng/mL and favorable intermediate-risk patients with ISUP grade 2 (<10% pattern 4), clinical stage T2b-c, PSA 10–20 ng/mL. Biopsies were performed transrectally or transperineally, at least 8 cores, targeted and systematic. Men had to be fit for curative therapy.

The exclusion criteria were a previous use of 5-alpha reductase inhibitors (5-ARIs) or any other therapy for PCa, as well as a history of allergic reactions to soy products. Table 1 shows the demographic characteristics of our population at baseline.

### 2.2. Study Design

The PRAEMUNE trial was an observational, retrospective, multicenter single-arm open-label trial with the administration of a fermented soy supplement for 18 months. Ethical approval was obtained in March 2020, by the ethical committee of the University Hospitals Leuven, Belgium (S62819). At inclusion, the following data were recorded: DRE, TRUS, PSA, PSA density (PSAD), MRI findings, pathology data with ISUP score, % pattern 4 and total biopsy tumor length. Trial visits were planned at 3, 6, 12 and 18 months after inclusion with DRE, PSA and TRUS. MRI and/or repeat biopsies were performed at the clinician’s discretion, consistent with the study results. All patients were given a nutritional supplement with isoflavones (10 mg) and fermented soy(equol) (60 mg), 1 capsule daily, and compliance or interruption was checked at each consultation.

### 2.3. Statistical Analysis

The primary outcome was PSA response defined as maximum PSA increase less than or equal to 0.87 ng/mL at month 12. This level of PSA increase was determined by taking the weighted average of 3 previous studies in similar populations in which PSA velocity (PSAV), expressed in ng/mL/year, was determined (Table 2) [13,14,15]. The group with PSA rise ≤ 0.87 ng/mL after 12 months was called “responders” (R), the other group “non-responders” (NR). Missing PSA values were imputed via a regression model using available PSA data at month 0, 3, 6, 12 and 18, as previously carried out by Ng et al. [13]. Hence, the primary and secondary outcomes using imputed data were also determined as sensitivity analyses. Secondary outcomes were disease progression based on unequivocal clinical progression and/or adverse histology on repeat biopsy (higher ISUP and/or increase % pattern 4), or switch to active therapy.

Analyses were conducted on an ‘intention-to-treat’ basis. Groups R and NR were compared via Mann–Whitney U or Chi-square tests where appropriate. Significance levels were set at 0.05.

## 3. Results

Of the 181 patients, 150 patients had an available PSA value at month 12. A waterfall plot clarifies the PSA evolution for the total cohort (without and with imputation) between baseline and 12 months (Figure 1A,B).

Primary outcome

For the primary outcome, 92/150 (61.3%) of patients showed a response in PSA level after one year, compared to 58/150 patients (38.7%). The imputed data showed that 101/181 (55.8%) patients were responders.

Secondary outcomes (Table 3)

For the comparison between R and NR, R were younger (*p* = 0.030) and showed lower PSA values at each timepoint. During the follow-up period, fewer MRIs were performed in the R group. Interestingly, the percentage of control biopsies taken at any time in the follow-up was lower in the R group (*p* = 0.052), and in the sensitivity analysis, this difference became statistically significant (*p* = 0.014) (See Appendix A). Moreover, the percentage of positive control biopsies taken at any time during follow-up was also lower in the R group (*p* = 0.151), and in the sensitivity analysis, this difference also became statistically significant (*p* = 0.048).

**Table 3 cancers-16-03634-t003:** Secondary outcomes per response group.

	Responders	Non-Responders	*p*-Value
Number (%)	92 (61.3%)	58 (38.7%)	
Median (IQR) age, years	66 (64–69)	70 (67–71)	0.030
MRI at baseline, n (%)	91 (98.9%)	54 (93.1%)	0.054
PIRADS (highest)			0.090
2	3 (3.3%)	0 (0.0%)	
3	16 (17.4%)	18 (33.3%)	
4	59 (65.2%)	28 (51.9%)	
5	13 (14.1%)	8 (14.8%)	
ISUP at baseline, n (%)			
ISUP 1	82 (89.1%)	45 (77.6%)	0.056
ISUP 2	10 (10.9%)	13 (22.4%)	
>10% Pattern 4 at baseline, n (%)	0 (0.0%)	0 (0.0%)	
Median (IQR) PSA density at baseline, ng/mL/cm^3^	0.14 (0.10–0.17)	0.14 (0.10–0.18)	0.558
Median (IQR) total biopsy tumor length at baseline, mm	5 (2–8)	5 (2–9)	0.966
Median (IQR) PSA at baseline, ng/mL	5.9 (5.5–6.4)	6.7 (5.7–8.1)	0.035
Median (IQR) PSA at 3 months, ng/mL	5.5 (4.9–6.2)	6.7 (6.1–8.2)	0.003
Median (IQR) PSA at 6 months, ng/mL	5.2 (4.8–6.0)	7.6 (6.7–8.6)	<0.001
Median (IQR) PSA at 12 months, ng/mL	5.4 (4.9–6.3)	8.4 (7.3–9.5)	<0.001
Median (IQR) PSA at 18 months, ng/mL	5.6 (4.9–6.9)	8.8 (7.0–10.2)	<0.001
Control MRI taken anytime, n (%)	79 (85.9%)	53 (91.4%)	0.127
PIRADS (highest)			0.743
2	3 (3.8%)	1 (1.9%)	
3	22 (27.9%)	13 (24.5%)	
4	45 (57.0%)	30 (56.6%)	
5	9 (11.4%)	9 (17.0%)	
Control biopsy taken anytime, n (%)	63 (68.5%)	48 (82.8%)	0.052
Control biopsy positive, n (%)	56 (60.9%)	43 (74.1%)	0.151
Control biopsy ISUP (n, %)			
ISUP 1	37 (40.2%)	23 (39.7%)	0.107
ISUP 2	19 (20.7%)	20 (34.5%)	
>10% Pattern 4 at control biopsy (n, %)	15 (16.3%)	12 (20.7%)	0.496
Median (IQR) Pattern 4% at control biopsy, %	30 (20–60)	35 (19–70)	0.792
Median (IQR) total biopsy tumor length at control biopsy, mm	11 (6–15)	10 (5–16)	0.559
Therapy initiated *, n (%)	8 (8.7%)	11 (19%)	0.117
Median (IQR) time until therapy, months	19 (16–21)	19 (14–20)	0.570

* Only therapy initiated at the 18-month follow-up timepoint (leniency period of 3 months) was taken into account.

At baseline, we see that in the R, the proportion of ISUP 1 predominates more than in the NR, and percentage-wise there are twice as many ISUP 2 in the NR. In control biopsies, the proportion of ISUP 2 increases markedly in both the R and NR, but in the R there are still twice as many ISUP1 versus ISUP 2, while in the NR the proportions of ISUP 1 and 2 are close to each other. According to the protocol, there were no patients with >10% pattern 4 at baseline, and we see in the control biopsies that the number of patients with >10% pattern 4 increased for both groups, and this pathology upgrade was slightly more pronounced in the NR (20.7%) versus the R (16.3%). There were no marked differences between R and NR regarding PSA density, as well as total biopsy tumor length at baseline and at control biopsy.

Finally, over the entire follow-up period, there was double the number of patients in whom therapy was initiated in the NR versus the R (not statistically significant), but there was no difference in median time until therapy.

Table 4 shows a detailed analysis by ISUP group for both R and NR, showing that more examinations (NMR/biopsy) have been carried out each time a comparison is made of ISUP 2 versus ISUP 1 (i.e., more aggressive tumors) and NR versus R (i.e., with a higher PSA risk profile).

A good safety profile was noticed for the nutritional supplement with no reported TRAEs.

## 4. Discussion

This study evaluated, based on possible PSA modulation, whether a potential effect could be identified during an 18-month challenge with a fermented soy supplement in the follow-up of patients under AS for low- and intermediate-risk PCa. First, we saw a clear difference between R and NR in number of MRIs performed, control biopsies and positive control biopsies where statistical significance was even reached between the two groups in the sensitivity analysis. Additionally, by performing a detailed analysis (Table 4), we saw that in this real-time study, more examinations (NMR/biopsy) were carried out in men with more aggressive tumors and men with a higher PSA risk profile. Thus, if there were fewer examinations in the study, it was mainly in men with lower risk profiles, making it unlikely that many (significant) tumors were missed in men who did not receive a control MRI or biopsy.

Secondly, there were proportionally more patients with an ISUP 1 represented in the R, which might indicate that a PSA modulation that could be attributed to a fermented soy supplement is less pronounced in more aggressive tumors. Thirdly, we observed a double percentage of initiated therapy in the NR in this study.

We try to argue that, on the one hand, obtaining PSA modulation may reduce the number of investigations in AS follow-up without likely missing significant PCa. On the other hand, patients who can obtain PSA modulation are more likely to have less aggressive tumors and may be more justified to remain under AS. Hypothetically, prescribing a fermented soy supplement could be considered to achieve PSA modulation and could have a possible impact on the follow-up of AS.

It has been shown that soy isoflavones, with genistein, daidzein and the metabolite equol, can exert a significant effect on PCa cells by interacting with multiple cellular mechanisms such as androgen- and estrogen-driven pathways, cell proliferation, angiogenesis, and metastasis, as well as through antioxidant and epigenetic properties [16,17].

An initial retrospective study showed a significant PSA modulating effect in patients at increased risk of PCa and negative prior biopsy after taking a fermented soy supplement for 6 months, and this was most pronounced in the subgroup of patients with a PSA > 4 ng/mL [11]. These results were confirmed in a prospective study where PSA modulation was achieved in 81% of included patients [12]. In this prospective study, there was a subsequent 2-year non-directive observational period which showed that obtaining PSA modulation had a clear impact on the number of examinations performed with less MRIs and fewer prostate biopsies. Finally, in patients who had achieved PSA stabilization, a significantly lower number of PCa and a lower number of ≥ISUP3 lesions were found in subsequent biopsies.

In addition to the focus on a fermented soy supplement, many diverse issues have already been studied that could potentially influence disease progression in AS. We mention lifestyle modifications (ranging from exercise and diet to coffee and vitamin supplements), medications (such as 5-ARIs and statins) and vaccines (PROSTVAC), as well as stronger anticancer drugs (Chlormadinone, fexapotide triflutate and androgen receptor-targeted agents (ARTAs)) [18,19,20,21,22,23,24,25,26].

Only for 5-ARIs, a significant reduction in PCa progression in AS could be observed (38% for Dutasteride versus 48% for placebo) without being associated with significant TRAEs [25]. For the ARTAs or so called next-generation androgen receptor inhibitors such as enzalutamide, apalutamide and darolutamide, even better results are obtained in terms of a reduction in PCa progression in AS, but this is accompanied by very significant TRAEs, even in up to 88% of patients in the case of enzalutamide [24,26]. Presumably for ARTAs, the benefit does not outweigh the potential harm, knowing that this medication is more likely to take its place in advanced (metastatic) PCa and that this medication is very expensive [21,22,23].

The role of PSA kinetics with PSAV and PSA doubling time (PSADT) is not clear as a possible parameter or predictor of progression of PCa. However, most guidelines agree that PSADT less than 3 years and PSAV more than 0.75 ng/mL/year should be considered unfavorable [27,28,29,30,31,32].

The percentage of patients in our study who proceeded to active therapy is slightly higher than what is reported in the literature, which can be explained on the one hand by the fact that there was a clear percentage of 15% of patients with ISUP 2 at inclusion, but mainly because in this real-world study both the investigator and the patient could participate in the decision [3].

The limitations of this study include the lack of a control group and the relatively short follow-up. Monitoring of adherence was not formal, but on the other hand, additional soy intake through the regular diet seems likely to be negligible in this real-world study. The bioavailability of soy in food is influenced by many factors and is generally assumed to be quite low [33]. As a result, the effect of soy in food is probably negligible compared to the concentrated administration in the food supplement. The protocol was non-directive, with decisions at each site left to the discretion of the investigator, and this potential variability between sites and investigators could influence the results. In addition, there was also the possibility that the patient could discontinue AS and switch to active therapy.

A prospective, randomized double-blind study is highly desirable to see whether these results can be confirmed. No TRAEs have been reported, with a good safety profile for dietary supplement.

## 5. Conclusions

This study examined the effect of a fermented soy supplement in patients under AS for low-risk and selected favorable intermediate-risk PCa. If a PSA modulation can be achieved, this is more pronounced in less aggressive tumors and appears to lead to a decrease in further investigations such as MRIs and biopsies, as well as a decrease in the number of positive biopsies. A fermented soy supplement may be useful in helping select patients who can remain under AS or who need to switch to active therapy.

## Figures and Tables

**Figure 1 cancers-16-03634-f001:**
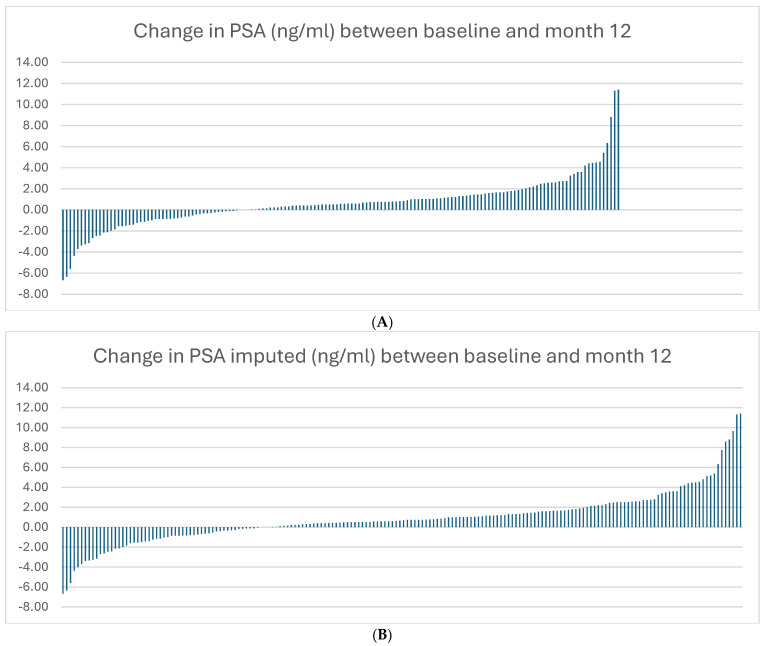
(**A**,**B**) Waterfall plot of PSA evolution for the total cohort between baseline and 12 months.

**Table 1 cancers-16-03634-t001:** Demographic characteristics of the entire population at baseline.

Characteristic	Value
Total included, n (%)	181
Participating centers, n (%)	4
Median (IQR) age, years	67 (62–72)
Median (IQR) PSA at baseline (PSA0), ng/mL	6.16 (4.55–8.34)
MRI at baseline, n (%)	176/181 (97.2%)
PIRADS (highest)	
2	7 (3.7%)
3	44 (24.3%)
4	108 (59.7%)
5	22 (12.3%)
ISUP at baseline, n (%)	
ISUP 1	153 (84.5%)
ISUP 2	28 (15.5%)
PSA density at baseline	
Mean (SD), ng/mL/cm^3^	0.141 (0.052)

**Table 2 cancers-16-03634-t002:** Characteristics of studies involving PSA velocity (PSAV).

Author	Year	Baseline PSA (ng/mL)	n	Age (Year)	Inclusion AS	PSAV (ng/mL/Year)	Follow-Up (Months)
NG et al. [13]	2008	Median: 6.4 (IQR 4.3–8.4)	199	Median: 66 (IQR 61–70)	PSA ≤ 15 ng/mL T_1_/T_2_a Gleason score ≤ 3 + 4 ≤50% + biopsy cores	Median: 0.71 (IQR 0.07–1.69)	24
Kotb et al. [14]	2010	Mean: 5.9 (range 0.18–20.4)	102	Mean: 66 (49–78)	Gleason ≤ 7	Mean: 0.88 (range 0–9.6)	Mean: 61 (24–172)
Fernandez- Anguita et al. [15]	2023	Mean: 8.27 (SD: 3.4)	86	Mean: 63.4 (SD: 6.4)	PSA ≤ 20 ng/mL T_1_/T_2_a Gleason ≤ 3 + 4 ≤3 + biopsy cores	Median: 1.3 (SD: 0.18)	Median: 32.3 (SD: 1.6)

n: number of patients in the study; PSAV: prostate-specific antigen velocity; IQR: interquartile range; SD: standard deviation.

**Table 4 cancers-16-03634-t004:** Detailed analysis by ISUP group for both responders (R) and non-responders (NR).

	ISUP (Baseline)			
	1		2	
	R (n = 82)	NR (n = 45)	R (n = 10)	NR (n = 13)
Control MRI taken anytime, n (%)	69 (84.1%)	41 (91.1%)	9 (90.0%)	13 (100.0%)
Control biopsy taken anytime, n (%)	56 (68.3%)	37 (82.2%)	7 (70.0%)	11 (84.6%)
Control biopsy positive, n (%)	49 (59.8%)	33 (73.3%)	7 (70.0%)	10 (76.9%)
>10% Pattern 4 at control biopsy n, (%)	10 (12.2%)	8 (17.8%)	5 (50.0%)	4 (30.8%)

## Data Availability

The original contributions presented in the study are included in the article/Appendix A; further inquiries can be directed to the corresponding author/s.

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
