# Peer review of "Nutritional Supplement with Fermented Soy in Patients Under Active Surveillance for Low-Risk or Intermediate-Risk Prostate Cancer: Results from the PRAEMUNE Trial"

_cancers, 2024, doi:10.3390/cancers16213634_

Reviewer 1 Report

Comments and Suggestions for Authors

This study is interesting but it should be better to have a prospective study with two groups of patients, one treated with fermented soy and one without.

Furthermore the follow-up is moderately short and I'm puzzled about the results at long term.  Which others drugs the patients enrolled take in ?

What about the tolerance of the nutritional supplement ?

The Authors should replay to this questions

Author Response

Response to Reviewers  

Reviewers' comments in italics: 

This study is interesting but it should be better to have a prospective study with two groups of patients, one treated with fermented soy and one without.

 We fully agree with this comment by the reviewer. It is also our explicit desire and planning to design a prospective and randomised study. Moreover, we mentioned this in the article at the very end under ‘discussion’: A prospective, randomised double-blind study is highly desirable to see if these results can be confirmed.

The reason why we could not do this at this time is the following. We are dealing with a nutritional supplement from a start-up new medical company that on the one hand believes in research and therefore uses fermented soy, but on the other hand did not yet have the resources to set up a larger randomised controlled trial. Therefore, we first started a retrospective study, then a prospective study and now this study in active surveillance (all studies mentioned in this article). We have now been able to demonstrate sufficient arguments to actually take the step towards a larger (international) randomised controlled trial for which negotiations are already ongoing and a protocol is already being worked on. Due to the favourable results of the previous studies, the company will now be able to provide the necessary support for the larger and more expensive study.

 Furthermore the follow-up is moderately short and I'm puzzled about the results at long term.  

 We certainly agree with this comment as well, and the results also surprised us somewhat. This makes us even more enthusiastic about going ahead with our planned randomised controlled trial with a fermented soy supplement, which I also referred to in the previous answer. Our hypothesis is that we can use a fermented soy supplement to sort of select between patients at low risk of aggressive cancer and others at higher risk, and hopefully our study results in the future can further substantiate this.

Which others drugs the patients enrolled take in ?

In terms of medication, we mainly looked at medication that can have a direct effect on the prostate and more specifically on the PSA level and therefore previous use of 5-alpha-reductase inhibitors or another therapy for prostate cancer was retained as exclusion criteria (listed in article under materials and methods/patients). Classical other medication could be continued (but no double use of other soy-containing products).

Regarding the control for additional dietary intake of soy, this may be an unmeasured (unmeasurable) confounder; however, in this real-world study, the outcomes could be more robust. Moreover, compared with the dietary supplement containing equol (which increases the bioavailability of isoflavones), the effect of dietary soy intake is likely to be negligible.

We have added an additional reference and sentence in the discussion section regarding varying additional dietary soy intake among subjects.

“The bioavailability of soy in food is influenced by many factors and is generally assumed to be quite low (Nielsen 2007). As a result, the effect of soy in food is probably negligible compared to the concentrated administration in the food supplement.”

What about the tolerance of the nutritional supplement ?

This was now the third study conducted with the same fermented soy food supplement, and no side effects or intolerances were ever reported. So we can say that the product is very well tolerated.

The Authors should replay to this questions

I hope we were able to provide a good clarification for the comments made and are always happy to address any further comments.

Reviewer 2 Report

Comments and Suggestions for Authors

The manuscript submitted by Hans Van Der Eecken and co-authors addressed the relevance of using a soya supplement as a strategy in the active surveillance of low-risk prostate cancer patients.

The manuscript is scientifically sound, well-written, and organized. The experimental plan and objectives are clearly presented, and the results and outcomes could improve the management of prostate cancer.

My general comment relies on the fact that neither the study design nor the discussion mentioned the individuals' dietary patterns. Was the dietary pattern analysed? Did the study include vegetarians or men with an essentially plant-based diet? How was this information considered? If not, something has to be discussed around this issue and included in the manuscript.  

Author Response

Response to reviewer 2

Response to Reviewers  

Reviewers' comments in italics: 

The manuscript submitted by Hans Van Der Eecken and co-authors addressed the relevance of using a soya supplement as a strategy in the active surveillance of low-risk prostate cancer patients.

The manuscript is scientifically sound, well-written, and organized. The experimental plan and objectives are clearly presented, and the results and outcomes could improve the management of prostate cancer.

My general comment relies on the fact that neither the study design nor the discussion mentioned the individuals' dietary patterns. Was the dietary pattern analysed? Did the study include vegetarians or men with an essentially plant-based diet? How was this information considered? If not, something has to be discussed around this issue and included in the manuscript.  

Thank you for pointing this out, we agree with this comment. The dietary pattern was not analysed. Regarding the control for additional dietary intake of soy, this may be an unmeasured (unmeasurable) confounder; however, in this real-world study, the outcomes could be more robust. Moreover, compared with the dietary supplement containing equol (which increases the bioavailability of isoflavones), the effect of dietary soy intake is likely to be negligible.

We have added an additional reference and sentence in the discussion section regarding varying additional dietary soy intake among subjects.

“The bioavailability of soy in food is influenced by many factors and is generally assumed to be quite low (Nielsen 2007). As a result, the effect of soy in food is probably negligible compared to the concentrated administration in the food supplement.”
